# A randomized trial of safety, acceptability and adherence of three rectal microbicide placebo formulations among young sexual and gender minorities who engage in receptive anal intercourse (MTN-035)

Jose A. Bauermeister[1]*, Clara Dominguez Islas[2,3], Yuqing Jiao[2,3], Ryan Tingler[1], Elizabeth Brown[2], Jillian Zemanek[2], Rebecca Giguere[4], Ivan Balan[4], Sherri Johnson[5], Nicole Macagna[5], Jonathan Lucas[5], Matthew Rose[5], Cindy Jacobson[6], Clare Collins[6], Edward Livant[6], Devika Singh[6], Ken Ho[7], Craig Hoesley[8], Albert Liu[9], Noel Kayange[10], Thesla Palanee-Phillips[11,12], Suwat Chariyalertsak[13], Pedro Gonzales[14], Jeanna Piper[15], on Behalf of the MTN-035 Protocol Team¶

1 Family and Community Health, University of Pennsylvania School of Nursing, Philadelphia, Pennsylvania, United States of America, 2 Statistical Center for HIV/AIDS Research & Prevention, Fred Hutchinson Cancer Center, Seattle, Washington, United States of America, 3 Vaccine and Infectious Disease Division, Fred Hutchinson Cancer Center, Seattle, Washington, United States of America, 4 Center for Translational Behavioral Science, Florida State University College of Medicine, Tallahassee, Florida, United States of America, 5 FHI 360, Durham, North Carolina, United States of America, 6 Magee-Women's Research Institute, Pittsburgh, Pennsylvania, United States of America, 7 Department of Medicine, University of Pittsburgh, Pittsburgh, Pennsylvania, United States of America, 8 University of Alabama at Birmingham, Birmingham, Alabama, United States of America, 9 San Francisco Department of Public Health, San Francisco, California, United States of America, 10 Johns Hopkins University Research Project, Blantyre, Malawi, 11 Wits Reproductive Health and HIV Institute, University of the Witwatersrand, School of Public Health, Johannesburg, South Africa, 12 Department of Epidemiology, School of Public Health, University of Washington, Seattle, Washington, United States of America, 13 Research Institute for Health Sciences, Faculty of Public Health, Chiang Mai University, Chiang Mai, Thailand, 14 IMPACTA, San Miguel, Peru, 15 Division of AIDS/NIAID/NIH, Bethesda, Maryland, United States of America

¶ Membership of the MTN-035 Protocol Team is provided in the Acknowledgments.
* bjose@nursing.upenn.edu

**Data Availability Statement:** Data cannot be shared publicly as some fields may include

## Abstract

Efforts to develop a range of HIV prevention products that can serve as behaviorally congruent viable alternatives to consistent condom use and oral pre-exposure prophylaxis (PrEP) remain crucial. MTN-035 was a randomized crossover trial seeking to evaluate the safety, acceptability, and adherence to three placebo modalities (insert, suppository, enema) prior to receptive anal intercourse (RAI). If participants had no RAI in a week, they were asked to use their assigned product without sex. We hypothesized that the modalities would be acceptable and safe for use prior to RAI, and that participants would report high adherence given their behavioral congruence with cleansing practices (e.g., douches and/or enemas) and their existing use to deliver medications (e.g., suppositories; fast-dissolving inserts) via the rectum. Participants (N = 217) were sexual and gender minorities enrolled in five different countries (Malawi, Peru, South Africa, Thailand, and the United States of America). Mean age was 24.9 years (range 18–35 years). 204 adverse events were reported by 98

identifiable Protected Health Information (PHI). Requests for de-identified datasets are available from the Microbicide Trials Network. A Dataset Request Form must be completed by the investigator requesting the data; the completed form must be then be submitted to the FHI 360 Clinical Research Manager (CRM) for the protocol. The data underlying the results presented in the study are available from the MTN. Data supporting the study findings were provided by the trial team of the Microbicide Trials Network (MTN) 035. Data are restricted and are not publicly available. Parties interested in accessing the MTN 035 data may contact the MTN Leadership Team at mtnadmmgr@mtnstopshiv.org for access to de-identified data.

**Funding:** The study was designed and implemented by the Microbicide Trials Network (MTN; https://mtnstopshiv.org/research/studies/mtn-035). From 2006 until November 30, 2021, the MTN was an HIV/AIDS clinical trial network funded by the National Institute of Allergy and Infectious Diseases (UM1AI068633, UM1AI068615, UM1AI106707), with co-funding from the Eunice Kennedy Shriver National Institute of Child Health and Human Development and the National Institute of Mental Health, all components of the U.S. National Institutes of Health. The content is solely the responsibility of the authors and does not necessarily represent the official views of the National Institutes of Health. NIH employees contributed to the study design, manuscript development and the decision to publish as well as providing safety oversight during study conduct but had no role in data collection and analysis.

**Competing interests:** AL has received funding for investigator sponsored research projects from Gilead Sciences and ViiV Healthcare. Gilead Sciences donated study drug to studies led by AL. This does not alter our adherence to PLOS ONE policies on sharing data and materials.

participants (45.2%); 37 (18.1%) were deemed related to the study products. The proportion of participants reporting "high acceptability" was 72% (95%CI: 65% - 78%) for inserts, 66% (95%CI: 59% - 73%) for suppositories, and 73% (95%CI: 66% - 79%) for enemas. The proportion of participants reporting fully adherent per protocol (i.e., at least one use per week) was 75% (95%CI: 69% - 81%) for inserts, 74% (95%CI: 68% - 80%) for suppositories, and 83% (95%CI: 77% - 88%) for enemas. Participants fully adherent per RAI-act was similar among the three products: insert (n = 99; 58.9%), suppository (n = 101; 58.0%) and enema (n = 107; 58.8%). The efficacy and effectiveness of emerging HIV prevention drug depends on safe and acceptable delivery modalities that are easy to use consistently. Our findings demonstrate the safety and acceptability of, and adherence to, enemas, inserts, and suppositories as potential modalities through which to deliver a rectal microbicide.

## Introduction

The HIV epidemic has affected people across the globe over the past four decades, with sexual and gender minorities (SGM; i.e., populations with same-sex or same-gender attractions or behaviors and who may identify with a non-heterosexual identity such as gay, bisexual, queer, etc.)carrying much of the burden of new diagnoses [1]. While HIV prevalence rates vary among SGM in different regions (4% sub-Saharan Africa, 32% South East Asia, 44% South America, and 55% North America), the risk of acquiring HIV is 28 times higher among men who have sex with men (MSM) than adult men and 14 times higher for transgender people when compared to adult women [1]. Innovative biomedical advancements across the HIV prevention continuum (e.g., HIV pre-exposure prophylaxis [PrEP]) have offered new opportunities to curtail HIV incidence [2], yet new HIV infections have remained high globally due to acceptability, access, uptake, and adherence challenges to these highly effective biomedical prevention tools [3]. Optimal uptake and adherence to PrEP has been hindered further by financial barriers and social stigma [4, 5], challenges accessing inclusive and sustained HIV prevention services [3, 6–8], or concerns regarding the possible side effects of systemic delivery of PrEP [9]. Therefore, efforts to develop a range of HIV prevention products that can serve as viable alternatives and/or complements to consistent condom use and oral PrEP, including formulations able to deliver multiple drugs (e.g., anti-HIV-1 and anti-HSV-2) in combination, remain crucial.

Researchers and advocates have proposed expanding modalities of PrEP delivery, including the use of rectal microbicides (RMs); topical biomedical products being developed to reduce the risk of HIV and other sexually transmitted infections (STIs) for use with sexual activity [9–11]. If found to be safe and effective, RMs may offer an episodic prevention modality for individuals who would perceive a pericoital prevention strategy more feasible and preferable to systemic daily or on demand oral PrEP modalities [9, 12]. In a randomized, cross-over trial comparing daily oral PrEP to two regimens (daily use and event-driven) of a rectal microbicide gel candidate, MSM and transgender women varied in their preferred strategy after using the products [13], with nearly 30% of the sample rating daily oral PrEP as the least preferred prevention method. Twenty percent of participants preferred an event-driven rectal microbicide strategy when engaging in condomless anal sex. Therefore, it is crucial for RMs to be designed so that they can be delivered via mechanisms that not only deliver enough drug to block HIV/STI transmission but are also a good behavioral fit with the intended end-users based on their lifestyles.

Most rectal microbicide candidates have been formulated as gels because of their similarities to lubricants, highlighting the potential for the rectal microbicide gel to be readily incorporated into users' sexual practices [10, 14–16]. While ideal from a behavioral congruence perspective, these gel formulations have required the use of an applicator to achieve sufficient drug delivery, which could present acceptability and adherence challenges for long-term uses [17, 18], At this time, it is unclear if topical gels would be able to deliver protective levels of HIV prevention drugs when used without an applicator [15, 19, 20]. As such, researchers and advocates have argued that other rectal delivery modalities should be considered, including suppositories, fast-dissolving inserts, and enemas [21–23].

Survey research with SGM populations across various countries has found high hypothetical acceptability to a rectal microbicide formulated as a suppository, insert, or enema [21, 24, 25]. while promising, the hypothetical nature of these studies limits our ability to understand whether SGM populations would find these modalities acceptable after using them with their partners prior to receptive anal intercourse (rai), and whether they would consistently use them when engaging in rai. to date, few studies with SGM populations have examined the acceptability, uptake of, and adherence to inserts, suppositories, or enemas as a rectal microbicide modality to be used prior to sex [26]. in a randomized, crossover acceptability trial where HIV-negative MSM used both 35 ml of placebo gel and an 8g placebo suppository prior to three rai occasions, respectively, participants noted moderate acceptability for the suppository, with the greatest proportion of participants preferring the gel over the suppository [26]. two clinical trials are examining the acceptability of rectal microbicide candidates as a fast-dissolving insert (mtn 039; nct04047420) and an enema (dream; nct04016233) for use among SGM populations, yet the results of these trials have yet to be released. moreover, no study has examined these modalities within the same trial, limiting our ability to compare their acceptability, safety, and adherence among SGM individuals who have used all three products rectally prior to rai. therefore, with the goal of supporting the development of behaviorally congruent rectal microbicide modalities for topical PrEP delivery, the Microbicide Trials Network (MTN) developed MTN-035 (DESIRE; Developing and Evaluating Short-Acting Innovations for Rectal Use) to identify acceptable modalities among SGM in five different countries: Malawi, Peru, South Africa, Thailand, and the United States of America.

The primary objectives of MTN-035 were to evaluate the acceptability and safety of and adherence to three placebo modalities–an insert, a suppository, and an enema–that could be used prior to RAI in a randomized, cross-over trial. We hypothesized that all three modalities would be acceptable and safe for use prior to RAI, and that SGM participants would report high adherence to these modalities given their behavioral congruence with cleansing practices (e.g., enemas) and their familiar use to deliver medications (e.g., suppositories; fast-dissolving inserts) via the rectum.

## Materials and methods

### Sample

HIV-uninfected transgender men, transgender women, and cisgender MSM between the ages of 18 and 35 were recruited into the trial (see Fig 1). Data collection took place between April 2019 and July 2020 in the United States (Pittsburgh, Pennsylvania; Birmingham, Alabama; and San Francisco, California), Thailand (Chiang Mai), Peru (Lima), Malawi (Blantyre), and South Africa (Johannesburg).

Participants were recruited from a variety of sources, including outpatient clinics, universities, community-based locations, online websites, and social networking applications. In addition, participants were also referred to the study from other local research projects, research

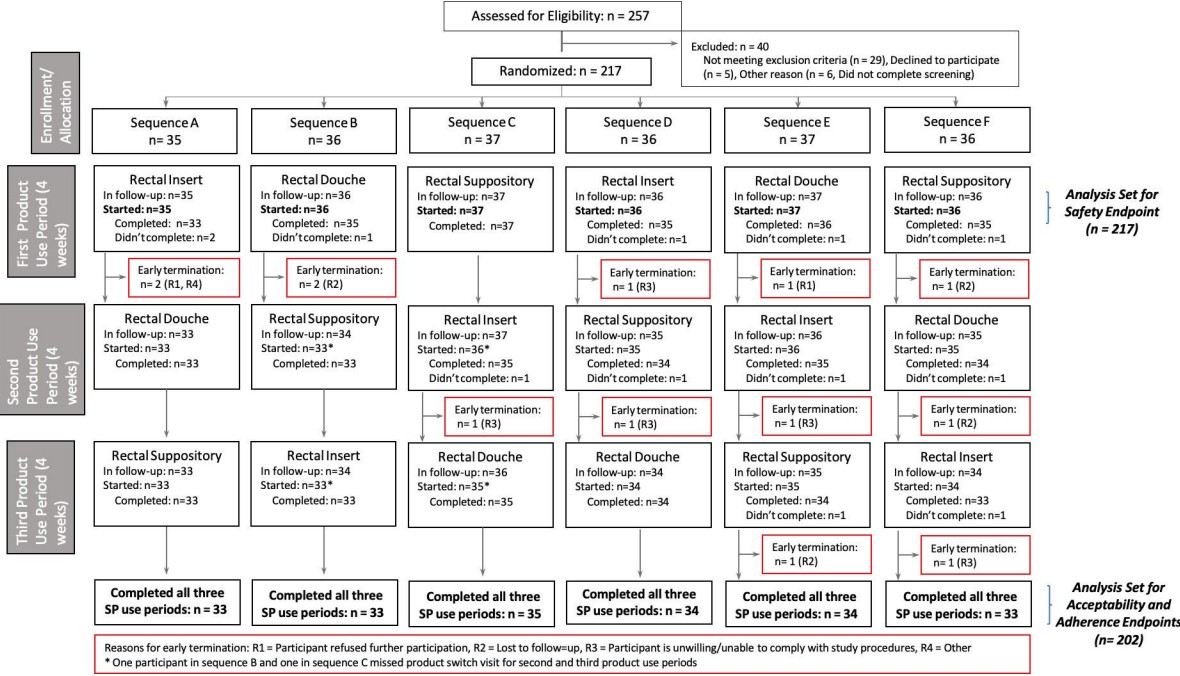

**Fig 1. CONSORT disposition of participants.**

registries and other health and social service providers. At some sites, prospective participants were pre-screened by phone, using an IRB-approved phone script, to assess presumptive eligibility based on select behavioral and medical eligibility requirements. This includes a review of the prospective participants' sexual history and engagement in receptive anal sex in their lifetime and within the previous three months. For those deemed presumably eligible when a phone screen was conducted, a screening visit was scheduled. A re- affirmation of all eligibility criteria was obtained and confirmed during a formal screening and enrollment visit, described below.

The study was reviewed and approved by Institutional Review Boards (IRB)/Ethics Committees at all participating institutions, including the Health Research Ethics Council, South African Health Products Regulatory Authority, the College of Medicine Research and Ethics Committee, the Johns Hopkins Bloomberg School of Public Health IRB, the Research Institute for Health Sciences Human Experimentation Committee, the Medical Device Control Division/Thai Food and Drug Administration (FDA), the IMPACTA Bioethics Committee, the Peruvian National Institute of Health (Instituto Nacional de Salud), the Peruvian FDA, the Peruvian Ministry of Health, the University of Pittsburgh IRB, the University of California San Francisco IRB, University of Alabama at Birmingham IRB, and the University of Pennsylvania IRB. This study was submitted to clincialtrials.gov on September 14, 2018, assigned number NCT03671239.

## Inclusion criteria

Inclusion criteria included: 1) men (cis or transgender) and transgender women between 18–35 years old; 2) ability and willingness to provide written informed consent in local language; 3) HIV- 1/2 uninfected at Screening and Enrollment; 4) ability and willingness to provide adequate contact and location information; 5) availability to return for all study visits and

willingness to comply with study participation requirements; 6) deemed to be in general good health by a healthcare provider at Screening and Enrollment; 7) a reported history of consensual RAI at least three times in the past three months and expectation to maintain at least that frequency of RAI during study participation; 8) willingness to not take part in other research studies involving drugs, medical devices, genital or rectal products, or vaccines for the duration of study participation; 9) For individuals who could get pregnant (transgender men with a female reproductive system), a negative pregnancy test at Screening and Enrollment; 10) For individuals who could get pregnant, use of an effective method of contraception for at least 30 days (inclusive) prior to Enrollment, and intention to use an effective method for the duration of study participation. Exclusion criteria included history of inflammatory bowel disease or anorectal condition impeding product placement or assessment of tolerability; anticipated use of non-study rectally administered products; any prior participation in research studies involving rectal products; having an active anorectal or reproductive tract infection requiring treatment or symptomatic urinary tract infection (these participants could be retested during screening and could enroll if resolved); and pregnancy or breast-feeding.

## Screening, enrollment and retention

Participants were screened for eligibility prior to enrolling in the study. All enrolled participants provided written informed consent. Participants returned to the clinic within the 45-day screening window where they completed administrative, behavioral, clinical, and laboratory procedures. Additionally, clinical results or treatments for urinary tract infections, genital/ reproductive tract infections, sexually transmitted infections (UTIs/RTIs/STIs) or other findings were provided as clinically indicated at all visits. At all clinic visits, participants were also dispensed condoms and lubricant. Consented and enrolled participants were then randomized into one of six sequences, each varying the order in which participants used the study placebo products, with a 1-week wash-out period between each 4-week product use period (Table 1).

Participants were randomly assigned in a 1:1:1:1:1:1 ratio to one of six study product application sequences (A-F), with the randomization configuration based on permuted blocks, to keep the allocation balanced. The randomization scheme, including enrollment of replacement participants, was generated and maintained by the MTN Statistical Data Manager Center (SCHARP), and it was configured in the Medidata Balance system prior to site activation. This allowed for participants being assigned to a randomized sequence by the system, only after site staff confirmed them as eligible and willing to enroll in the study.

Each participant was followed for approximately 3.5 months and was expected to complete eight visits (including Screening and Enrollment visits). A regular visit was considered missed if the participant did not complete any part of the visit within the visit window. If an interim visit was completed to make up for the missed regular visit, then the missed regular visit was calculated as completed.

**Table 1. Randomization sequence order.**

| Sequence | Period 1 (4 weeks) | Washout period (~1 week) | Period 2 (4 weeks) | Washout period (~1 week) | Period 3 (4 weeks) |
|---|---|---|---|---|---|
| A | Rectal Insert | | Rectal Enema | | Rectal Suppository |
| B | Rectal Enema | | Rectal Suppository | | Rectal Insert |
| C | Rectal Suppository | | Rectal Insert | | Rectal Enema |
| D | Rectal Insert | | Rectal Suppository | | Rectal Enema |
| E | Rectal Enema | | Rectal Insert | | Rectal Suppository |
| F | Rectal Suppository | | Rectal Enema | | Rectal Insert |

## Study procedures

Each participant received placebo inserts, placebo suppositories, and placebo (water) enema bottles for pericoital rectal administration (see Fig 2). The rectal suppository is approximately 3–3.8 cm (1.2–1.5 inches) long and 2 grams in weight. The placebo rectal suppository consists of a Witepsol® H5 (IOI Oleochemical) base and contains 15% diglyceride and not more than 1% monoglyceride content. The placebo rectal insert provided by CONRAD is formulated into white to off-white uncoated solid dosage forms in a bullet shape. The insert contains the following inactive excipients: isomalt, xylitol, sodium CMC, povidone, hydroxypropyl methylcellulose, poloxamer 188, sodium stearyl fumarate and magnesium stearate. The insert is 1.5 cm (0.6 inches) long, 0.7 cm (0.28 inches) wide, 0.6 cm (0.23 inches) in height, and approximately 500 mg in weight.

The products were administered in order of the assigned sequence and prior to each respective product use period. Participants were instructed to use one dose of the assigned study product between 30 minutes and 3 hours prior to RAI, following their usual pre-RAI practices, and not to use more than one product dose in 24 hours. If a participant did not engage in RAI in a given week, they were asked to insert a dose of the product in the absence of RAI. Participants self-administered the first dose of each product in the clinic to ensure correct administration.

The schedule of participants' study activities is depicted in Fig 3. At Visit 2, participants were provided with their first rectal product for period 1, based on their assigned sequence. For Visits 3, 5, and 7, participants returned to the clinic for the product use end visits. At these visits, participants completed study procedures, including pharyngeal, urine, blood, pelvic (individuals with a vagina or neovagina), and anorectal tests, if indicated (required at Visit 7). Participants completed a baseline computer assisted self-interview (CASI) during their enrollment visit (Visit 2) and at the end of each product use end visit (Visits 3, 5, and 7).

**Insert**

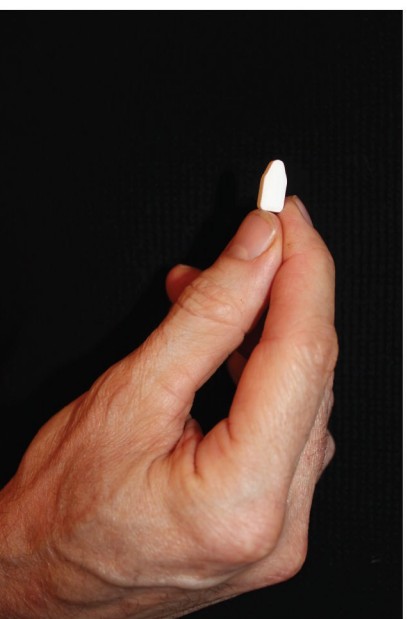

**Enema**

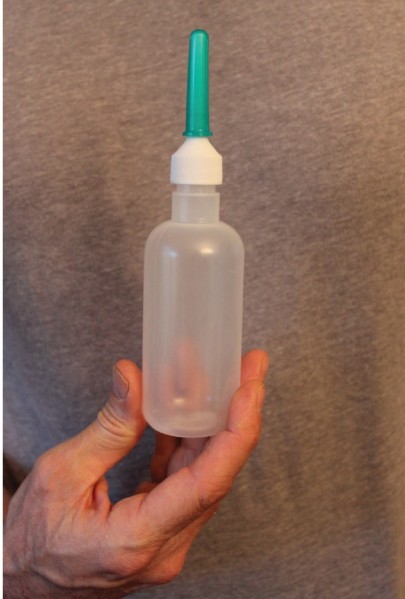

**Suppository**

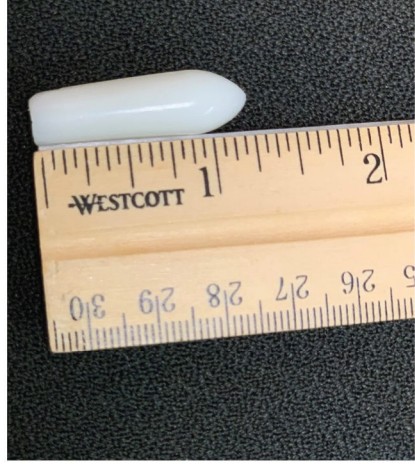

**Fig 2. MTN-035 placebo study products.**

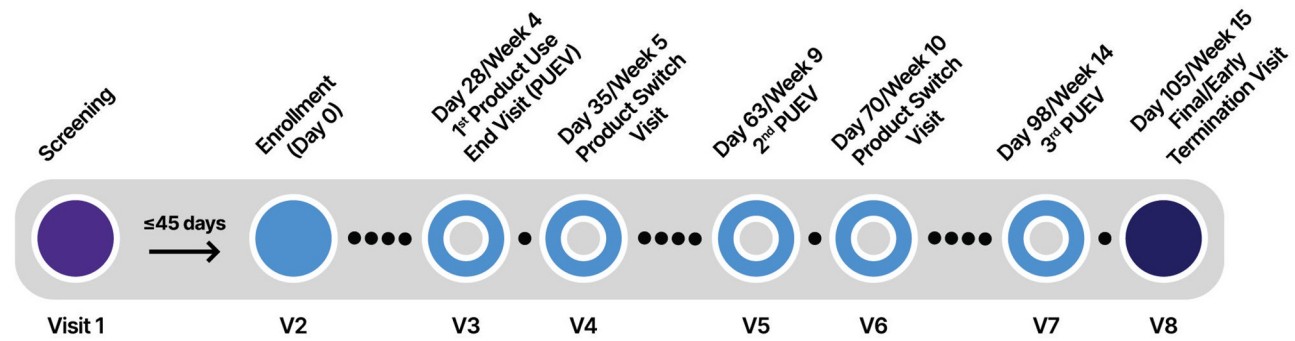

**Fig 3. MTN-035 study schema.**

After an approximately 7-day washout period following study product use periods, participants returned to the clinic to complete Visits 4 and 6. At these visits, participants completed study procedures, including pharyngeal, urine, blood, pelvic (individuals with a vagina or neo-vagina), and anorectal tests, if indicated. Additionally, participants self-administered one dose of the product they were dispensed and collected the remaining product in their sequence to use for the next four weeks during periods 2 and 3; they were also given product use instructions.

Visit 8 served as the follow-up safety contact and termination visit where participants completed study procedures as well as received clinical results or treatment for UTIs/RTIs/STIs or other findings. Participant reimbursement was based on local guidelines and approved by the local IRBs/ECs prior to study implementation.

We undertook several efforts to minimize the impact of the COVID-19 pandemic during our data collection period, including revising when and how products were dispensed during periods of COVID-19 restrictions. Of the 78 enrolled participants when the COVID-19 pandemic began, only four did not receive all three products (see Jacobson et al. [27] for details).

## Primary safety endpoint

Our primary safety endpoint was defined as the presence of a Grade 2 or higher related adverse events (AEs) as defined by the Division of AIDS Table for Grading the Severity of Adult and Pediatric Adverse Events, Corrected Version 2.1, July 2017 and Addenda 1, 2 and 3 (Female Genital [Dated November 2007], Male Genital [Dated November 2007] and Rectal [Clarification dated May 2012] Grading Tables for Use in Microbicide Studies [28–31].

## Primary acceptability endpoint

Acceptability endpoints were based on participants' responses to the CASI for each product at their respective product use end visits (Visits 3, 5 and 7). Using a 10-point scale (1 = Very Unlikely; 10 = Very Likely), participants were asked to answer the following question about their most recently used product: "Think about the positive and negative experiences you have had using the [study product] during the past 4-week period. If this [study product] was available and it provided some protection against HIV, how likely would you be to use it before receptive anal sex?". The endpoint was operationalized as binary, with scores 1 to 6 grouped as "low acceptability" and scores 7 to 10 as "high acceptability".

## Primary adherence endpoints

Adherence to use of each assigned product, as per-protocol, was based on the number of weeks that a participant missed using an assigned product (0 to 4 weeks), using a given product use end visit CASI assessment: "The following questions refer to your use of the study provided rectal [study product: enema, insert or suppository] over the past 4 weeks. You were asked to insert the study provided rectal [study product] in your rectum at least once a week during the past 4 weeks. However, for different reasons, people might have encountered difficulties using the [study product]. Thinking about your experience during these past four weeks, in how many of the weeks did you miss a rectal [study product] application?". The adherence per-protocol was operationalized as binary, with participants who reported not having missed any application classified as "adherent".

Additionally, adherence per RAI-act was defined (for participant-periods with at least one RAI act was reported) as the proportion of times that participants reported having used the study-provided enema, insert or suppository before RAI. We determined adherence per RAI act by dividing participants' responses to their CASI assessments regarding the number of times a participant noted using the study product before RAI by the total number of RAI acts self-reported over the same 4-week period. Participants are classified as fully adherent per RAI-act if they reported using the study product for all reported RAI acts.

## Statistical analysis

There is no control group for comparison in this study. The main goal was not to compare between the three different placebo modalities, but to obtain overall rates of acceptability, adherence, and safety of each modality. The selected sample size provides at least 90% power to rule out rates of acceptability or adherence below 70%.

Baseline characteristics are described for all enrolled participants. The study's safety endpoint was evaluated among participants who received at least one of the study products (excluding any study periods when participants did not receive a product). The acceptability and adherence primary endpoints were evaluated among participants who received all three study products and completed the scheduled product use (which we refer to as the "per-protocol" subset), thus providing relevant data from all three periods of study. To support the generalizability of acceptability and adherence results, we compared baseline characteristics between participants in the per-protocol subset and those who were lost to follow-up.

For the safety endpoint, the proportion of participants with Grade 2 or higher related AEs is reported, along with a 95% confidence interval (CI) (Clopper-Pearson method). For adherence and acceptability endpoints, the proportion of participants classified as adherent, or who reported a high acceptability score, are also provided, along with 95% CIs.

To test for potential effects of the assigned sequence and/or sites in the overall acceptability and adherence, generalized linear mixed models with a logistic link function were used. The participant's assigned sequence, site, and the modality used in the study period were included as fixed effects, with a random effect at the participant level to account for the cross-over design. To test if overall differences exist between sites or assigned sequences, omnibus likelihood ratio tests were used. No adjustment of p-values was performed.

## Results

### Demographics of enrolled participants

We screened 257 individuals across the seven study sites and enrolled 217 participants in five countries; 40 were not enrolled: 29 were not eligible, five were eligible but did not enroll, and

six did not complete their screening (see Fig 1). Overall, the retention rates were above 90% for all study visits.

The mean age was 24.9 years (SD = 4.7), ranging from 18 to 35 years old. Most of the sample reported having a male sex indicator assigned at birth (n = 214; 99%). Twenty percent of the sample identified as a gender minority. Overall, 13 (14%) participants in the U.S sites identified as Hispanic/Latinx. The racial, ethnic, and tribal affiliation of participants across the study sites is noted in Table 2.

Participants reported having had an average of 3 male partners (SD = 6.35; range: 0–70) in the prior 30 days. Participants' average total number of RAI occasions during that 30-day period was 4.76 (SD = 8.50; range: 0–100), with an average of 2.57 (SD = 7.74; range: 0–100) condomless RAI occasions self-reported during the same period. Two thirds of participants (n = 142; 65.4%) reported prior use of an enema, with fewer participants self-reporting that they had used a suppository (n = 8; 3.7%) or insert (n = 10; 4.6%) prior to RAI.

**Table 2. Participants' sociodemographic characteristics, overall and by site.**

| | All Sites (N = 217) | Birmingham, USA (N = 33) | Pittsburgh, USA (N = 33) | San Francisco, USA (N = 30) | Blantyre, Malawi (N = 31) | Chiang Mai, Thailand (N = 30) | Johannesburg, South Africa (N = 30) | Lima, Peru (N = 30) |
|---|---|---|---|---|---|---|---|---|
| Age (years), M(SD) | 24.9 (4.7) | 25.7 (5.1) | 25.5 (4.8) | 28.6 (3.9) | 24.6 (4.6) | 23.3 (3.3) | 21.9 (3.0) | 24.7 (4.7) |
| Sex Assigned at Birth, N (%) | | | | | | | | |
| Male | 214 (99%) | 31 (94%) | 32 (97%) | 30 (100%) | 31 (100%) | 30 (100%) | 30 (100%) | 30 (100%) |
| Female | 3 (1%) | 2 (6%) | 1 (3%) | 0 (0%) | 0 (0%) | 0 (0%) | 0 (0%) | 0 (0%) |
| Gender | | | | | | | | |
| Male | 173 (80%) | 28 (85%) | 28 (85%) | 27 (90%) | 19 (61%) | 22 (73%) | 28 (93%) | 21 (70%) |
| Female | 2 (1%) | 0 (0%) | 0 (0%) | 1 (3%) | 0 (0%) | 0 (0%) | 0 (0%) | 1 (3%) |
| Transgender Male | 2 (1%) | 1 (3%) | 1 (3%) | 0 (0%) | 0 (0%) | 0 (0%) | 0 (0%) | 0 (0%) |
| Transgender Female | 19 (9%) | 2 (6%) | 0 (0%) | 0 (0%) | 0 (0%) | 8 (27%) | 2 (7%) | 7 (23%) |
| Gender Nonconforming/ Variant | 5 (2%) | 0 (0%) | 3 (9%) | 1 (3%) | 0 (0%) | 0 (0%) | 0 (0%) | 1 (3%) |
| Other Gender | 10 (5%) | 0 (0%) | 0 (0%) | 1 (3%) | 9 (29%) | 0 (0%) | 0 (0%) | 0 (0%) |
| Multiple Genders | 6 (3%) | 2 (6%) | 1 (3%) | 0 (0%) | 3 (10%) | 0 (0%) | 0 (0%) | 0 (0%) |
| Latinx Ethnicity (U.S. Sites) | 13 (6%) | 2 (6%) | 3 (9%) | 8 (27%) | 0 (0%) | 0 (0%) | 0 (0%) | 0 (0%) |
| Race (U.S. Sites) or Ethnic Group (non-U.S. sites) | | | | | | | | |
| Asian | 8 (4%) | 0 (0%) | 2 (6%) | 6 (20%) | 0 (0%) | 0 (0%) | 0 (0%) | 0 (0%) |
| Black or African American | 16 (7%) | 13 (39%) | 1 (3%) | 2 (7%) | 0 (0%) | 0 (0%) | 0 (0%) | 0 (0%) |
| Native Hawaiian or Other Pacific Islander | 1 (1%) | 1 (3%) | 0 (%) | 0 (0%) | 0 (0%) | 0 (0%) | 0 (0%) | 0 (0%) |
| White | 66 (30%) | 19 (58%) | 28 (85%) | 19 (63%) | 0 (0%) | 0 (0%) | 0 (0%) | 0 (0%) |
| Multiple Races | 33 (15%) | 0 (0%) | 2 (6%) | 1 (3%) | 0 (0%) | 0 (0%) | 0 (0%) | 30 (100%) |
| Thai | 30 (14%) | 0 (0%) | 0 (0%) | 0 (0%) | 0 (0%) | 30 (100%) | 0 (0%) | 0 (0%) |
| Xhosa | 5 (2%) | 0 (0%) | 0 (0%) | 0 (0%) | 0 (0%) | 0 (0%) | 5 (17%) | 0 (0%) |
| Zulu | 14 (6%) | 0 (0%) | 0 (0%) | 0 (0%) | 0 (0%) | 0 (0%) | 14 (47%) | 0 (0%) |
| Other African Ethnic Group | 38 (18%) | 0 (0%) | 0 (0%) | 0 (0%) | 31 (100%) | 0 (0%) | 7 (23%) | 0 (0%) |
| Other | 6 (3%) | 0 (0%) | 0 (0%) | 2 (7%) | 0 (0%) | 0 (0%) | 4 (13%) | 0 (0%) |

USA: United States of America

## Study product use period completion

A study product use period was considered completed if the participant received the study product and completed the scheduled study product use period. Study product use period completion rates were 94% for the insert, 95% for the suppository, and 95% for the enema. Among enrolled participants, 92% exited the study at their scheduled end of study visit. Reasons for early study terminations included "Participant refused further participation" [n = 2 (1%)], "Participant is unwilling/unable to comply with required procedures" [n = 6 (3%)], "Lost to follow-up" [n = 4 (2%)], "Investigator decision" [n = 1 (<1%)], "Unable to contact participant" [n = 3 (1%)], "HIV infection" [n = 1 (<1%)], and "Other, specify" [n = 1 (<1%), participant relocating].

The per-protocol subset of participants (those who completed all three study product use periods) included 202 (93%) out of the 217 participants enrolled (see Fig 1). No major differences were observed in the baseline characteristics of these participants (site, randomization sequence, age, sex assigned at birth or gender) when compared to those of participants not included in the per-protocol subset.

## Primary safety endpoint

There were 204 AEs in the study reported by 98 participants (45% of the total sample; see Table 3). One hundred sixty-seven (81.9%) were classified as not related to the study products, and the remaining 37 (18.1%) were classified as related to the study products and occurred in 24 (11.1%) of participants. Thirty-five product-related AEs were graded as mild; two were graded as moderate (our primary safety endpoint) and both occurred during periods of insert use. Product-related AEs included abdominal distention (n = 1), abdominal pain (n = 5), anal pruritus (graded as moderate, n = 1), anorectal discomfort (n = 7), constipation (n = 1), defecation urgency (n = 3), diarrhoea (n = 4), dyschezia (n = 1), flatulence (n = 6), nausea (n = 1), rectal haemorrhage (n = 1), rectal tenesmus (n = 4), and malaise (n = 2, one graded as moderate). No events were graded as potentially life-threatening or resulting in death. There were no pregnancies reported by participants in this study. Two participants tested positive for HIV infection during follow-up while enrolled in the study.

## Primary acceptability endpoint

The proportion of participants reporting "high acceptability" was for inserts: 72% (95%CI: 65% - 78%), suppositories: 66% (95%CI: 59% - 73%), and enemas: 73% (95%CI: 66% - 79%) (see Table 4).

From the logistic mixed model (see Table 5), no statistically significant differences in product acceptability were observed between products. Participants in Birmingham, Blantyre,

**Table 3. Number of adverse events (AEs) reported.**

|  | Total | Not Related | Related |
|---|---|---|---|
| Severity Grade | n (%) | n (%) | n (%) |
| Grade 1 Mild | 95 (46.6%) | 60 (63.2%) | 35 (36.8%) |
| Grade 2 Moderate | 107 (52.5%) | 105 (98.1%) | 2 (1.9%) |
| Grade 3 Severe | 2 (0.9%) | 2 (100%) | 0 (0%) |
| Grade 4 Potentially Life-Threatening | 0 (0%) | 0 (0%) | 0 (0%) |
| Grade 5 Death | 0 (0%) | 0 (0%) | 0 (0%) |
| Total | 204 (100%) | 167 (81.9%) | 37 (18.1%) |

Notes. 98 out of the 217 participants reported one or more AEs.

**Table 4. Acceptability and adherence by study product.**

| | Acceptability | | | Adherence | | |
|---|---|---|---|---|---|---|
| | Rectal Insert | Suppository | Rectal Enema | Rectal Insert | Suppository | Rectal Enema |
| Percentage with High Acceptability/Adherence and 95% Confidence Interval | 72% (65%, 78%) | 66% (59%, 73%) | 73% (66%, 79%) | 75% (69%, 81%) | 74% (68%, 80%) | 83% (77%, 88%) |
| Participants with high acceptability/adherence by site | | | | | | |
| Birmingham, USA | 21/29 (72%) | 19/29 (66%) | 23/29 (79%) | 26/29 (90%) | 22/29 (76%) | 24/29 (83%) |
| Blantyre, Malawi | 24/29 (83%) | 21/29 (72%) | 26/29 (90%) | 23/29 (79%) | 16/29 (55%) | 20/29 (69%) |
| Chiang Mai, Thailand | 25/30 (83%) | 21/30 (70%) | 13/30 (43%) | 25/30 (83%) | 30/30 (100%) | 29/30 (97%) |
| Johannesburg, South Africa | 22/27 (81%) | 21/27 (78%) | 20/28 (71%) | 10/27 (37%) | 14/27 (52%) | 18/28 (64%) |
| Lima, Peru | 18/26 (69%) | 18/26 (69%) | 23/25 (92%) | 22/26 (85%) | 19/26 (73%) | 22/25 (88%) |
| Pittsburgh, USA | 17/30 (57%) | 16/30 (53%) | 23/30 (77%) | 20/30 (67%) | 23/30 (77%) | 26/30 (87%) |
| San Francisco, USA | 16/30 (53%) | 16/30 (53%) | 15/30 (50%) | 24/30 (80%) | 24/30 (80%) | 26/30 (87%) |

Notes. Estimates exclude missing data for acceptability (Rectal insert (n = 3); Suppository (n = 2); Enema (n = 5)) and adherence (Rectal insert (n = 3); Suppository (n = 3); Enema (n = 2)).

Johannesburg, and Lima being more likely (between 2.74 and 6.42 times more likely) to report high acceptability than participants in San Francisco. No significant differences were observed by product sequence.

## Primary adherence endpoint

The adherence primary endpoint was evaluated on the per-protocol subset of 202 participants (see Table 4). The proportion of participants reporting full adherence per protocol was: inserts 75% (95%CI: 69% - 81%), suppositories: 74% (95%CI: 68% - 80%), and enemas: 83% (95%CI: 77% - 88%).

As noted in Table 5, the observed differences in adherence per protocol across products were statistically significant. Statistically significant differences were also observed between some sites, with participants in the Johannesburg site were also less likely to be fully adherent when compared to their peers in San Francisco. Assigned product sequence was not associated with adherence.

## Adherence per-sex-act

Participants in the per-protocol subset reported an average of about 7 sex acts in the 4-week period of product use: insert (M = 7.2; SE = 0.7), suppository (M = 7.7; SE = 0.7), and enema (M = 7.8; SE = 0.8). Among participants who reported at least one sex act during the product use period, the percentages of participants fully adherent per RAI-act were similar among the three study products: insert (n = 99/168; 58.9%), suppository (n = 101/174; 58.0%) and enema (n = 107/182; 58.8%).

## Discussion

RMs are needed for individuals with an increased chance of acquiring HIV through RAI, particularly young SGM across the globe [23]. Given ongoing challenges to ensure equitable uptake and adherence to systemic oral PrEP and the desire for a diverse suite of HIV prevention products, it is important to expand the HIV/STI prevention pipeline by exploring whether different types of products that could be delivered through various modalities are safe, acceptable and adherable, and behaviorally congruent with the intended end-users' RAI practices [9].

**Table 5. Results from a linear mixed models with logistic link function estimating the odds ratio of reporting High (A) Acceptability to Study Product and (B) Adherence to Study product, for the Rectal Insert and Rectal Suppository, relative to the Rectal Enema, after accounting for site and sequence order.**

|  | Acceptability | | | | Adherence | | | |
|---|---|---|---|---|---|---|---|---|
|  | Odds Ratio Estimate | Lower CI | Upper CI | p-value | Odds Ratio Estimate | Lower CI | Upper CI | p-value |
| Product |  |  |  |  |  |  |  |  |
| Enema | REF | REF | REF | REF | REF | REF | REF | REF |
| Insert | .97 | .60 | 1.57 | .90 | .51 | .28 | .94 | .03 |
| Suppository | .69 | .43 | 1.11 | .13 | .49 | .26 | .89 | .02 |
| Site |  |  |  |  |  |  |  |  |
| San Francisco, USA | REF | REF | REF | REF | REF | REF | REF | REF |
| Birmingham, USA | 2.74 | 1.24 | 6.07 | .01 | 1.06 | .28 | 3.96 | .93 |
| Blantyre, Malawi | 6.42 | 2.60 | 15.83 | < .001 | .33 | .09 | 1.16 | .08 |
| Chiang Mai, Thailand | 1.94 | .90 | 4.17 | .09 | 3.52 | .83 | 14.97 | .09 |
| Johannesburg, South Africa | 4.36 | 1.85 | 10.29 | .001 | .10 | .03 | .38 | .001 |
| Lima, Peru | 3.47 | 1.49 | 8.06 | .004 | .89 | .23 | 3.38 | .86 |
| Pittsburgh, USA | 1.65 | .77 | 3.52 | .20 | .67 | .19 | 2.41 | .54 |
| Sequence Order |  |  |  |  |  |  |  |  |
| A | REF | REF | REF | REF | REF | REF | REF | REF |
| B | 1.52 | .68 | 3.39 | .31 | .99 | .30 | 3.32 | .99 |
| C | .80 | .38 | 1.72 | .57 | 1.25 | .38 | 4.12 | .71 |
| D | .91 | .42 | 1.96 | .81 | .79 | .24 | 2.56 | .69 |
| E | 1.01 | .47 | 2.18 | .98 | 1.02 | .31 | 3.35 | .97 |
| F | .91 | .42 | 1.97 | .80 | 1.50 | .44 | 5.10 | .51 |

*Note.* Rectal enema, the San Francisco site, and sequence A are used as reference levels for study product, site, and randomized sequence, respectively.

All three administration modalities with placebo products were found to be safe for use, with less than 20% of AEs deemed related to the study products/modalities and only two related events being graded as having moderate clinical severity or higher. These findings align with our study hypothesis. The safety profile of all three administration modalities with placebo products is promising and underscores the potential of all three modalities as viable for rectal drug delivery. The low incidence of distinct product-related AEs (e.g., diarrhea, flatulence) across study products is also noteworthy, as it mirrors common AEs reported when using over-the-counter rectal products (e.g., enemas) and prior rectal microbicide candidates in clinical trials (e.g., gels). while we should continue to reduce the occurrence of any AEs during product development, the promising safety profiles for the three modalities using placebo products employed in this trial offer a threshold whereby future clinical trials examining these same modalities with active drug ingredients can benchmark their safety endpoints.

Consistent with our hypotheses for our acceptability and adherence endpoints, SGM participants reported high overall acceptability for all three products and high overall adherence per protocol and per RAI act. While survey research literature has noted high hypothetical acceptability to RMs as an HIV prevention strategy prior to RAI [9, 14, 15, 17, 20, 24, 32], there has been limited research exploring the real-world acceptability and adherence of these three products. The absence of these data is troubling from a product development perspective given the increasingly restrictive costs and resources required to empirically evaluate a promising biomedical HIV prevention candidate. In the absence of product acceptability and adherence data, the effectiveness of a PrEP candidate may be compromised if the intended end-users are not willing to use it. Future research examining the acceptability of and adherence to diverse PrEP candidates should remain a priority in clinical trials.

While overall acceptability and adherence were high for each product, we observed differences in acceptability and adherence after SGM participants had used all three modalities. While the products were rated similarly in their acceptability and there were no differences based on participants' assigned product sequence, adherence to the enema was higher when compared to the fast-dissolving insert or the suppository. Given the high prevalence of rectal douching prior to RAI among SGM globally [33], the greater adherence to the enema may be indicative of congruence between the benefits afforded by the modality and users' behavioral practices prior to RAI. In a recent study, for instance, Carballo-Diéguez and colleagues [21] found that over 80% of SGM participants reported douching prior to RAI to rinse their rectum and feel clean, avoid smelling bad, and/or enhance their sexual pleasure. Given the limited data on the use of inserts or suppositories by SGM populations, however, it is unclear if these products could also yield behaviorally congruent benefits prior to or during RAI, including increasing sexual pleasure and lubricity during sex, and result in greater adherence in future studies.

We also observed differences for both acceptability and adherence between participants living in the different communities across the five countries participating in the trial. Compared to San Francisco, participants in the other regions reported greater acceptability of the three rectal modalities under study. Consistent with prior research with hypothetical and real-world studies with rectal candidates [9, 13, 26, 34], these regional variations suggest that some modalities may be more acceptable or adherable in some contexts than others. It is possible that this difference in acceptability may be related to the various efficacious PrEP modalities already available in San Francisco (e.g., daily, and event-driven oral PrEP) and which may not be available or accessible in other regions. As such, participants in the San Francisco site may weigh acceptability differently given their ability to compare these placebo modalities against efficacious PrEP technologies. We also found differential adherence between participants in the San Francisco and the Johannesburg sites; however, it is unclear what may have contributed to these differences. Although the collected data suggests differences by sites, future research, both qualitative and quantitative, may be warranted given the number of comparisons and the absence of clear hypotheses to understand the factors contributing to these differences.

## Strengths & limitations

This study had several strengths. First, this is the first study to examine the safety and acceptability of and adherence to these three promising modalities for rectal drug delivery prior to RAI. Examining each product's use in real life contexts strengthens the social validity of our findings and the potential use for these three modes of delivery in the future. Second, the crossover randomized design of our trial allowed us to assess SGM participants' acceptability of and adherence to these three modalities within the same trial, offering a unique opportunity to compare their acceptability, safety, and adherence within individuals who used all three products prior to RAI. Third, given the variability in both legal protections and social acceptance of SGM people between and within these countries, our ability to recruit and retain a large sample of young SGM living in geographically and socio-politically diverse countries is noteworthy and strengthens the generalizability of our findings to diverse contexts. Finally, our efforts to minimize the impact of the COVID-19 pandemic during our data collection period minimized interruptions to our trial and ensured that rigor was preserved [27].

Nonetheless, our trial also had several limitations. First, self-reported responses tend to be favorable due to social desirability; however, we tried to minimize bias by having participants complete their questionnaires in a private location during their clinical visits. Second, there is a possibility of recall bias when participants completed their surveys. Third, we recruited a

convenience sample of participants willing to use each of the study products at least once per week, as required by the protocol. We acknowledge that the generalizability of our clinical trial findings may not be representative of all individuals practicing RAI. Fourth, given the placebo nature of the three products used in our trial, we were unable to employ a biological confirmation method regarding participants' product use and adherence, or examine the extent and duration of rectal coverage afforded by each product during and after sex. These data will be crucial in the future, particularly as drug candidates are embedded into these modalities and tested for safety and adherence. Fifth, we were unable to recruit the sexual partners of our study participants. Given existing data regarding partners' roles in young SGMs' decision-making when selecting HIV prevention strategies prior to sex, future research examining these dyadic dynamics may be warranted. Finally, while we designed our clinical trial to resemble participants' product use to as close as 'real-world' settings possible while maintaining rigor, we acknowledge that the trial protocols may hinder the social validity of the findings. Future research examining these products in real-world situations may further clarify their potential for use as rectal microbicide modalities.

## Conclusions

Advances in biomedical strategies for HIV prevention continue to emerge. Efforts to diversify HIV prevention options will strengthen our ability to reduce new HIV infections among SGM, whether some SGM desire systemic modalities (e.g., daily oral PrEP; PrEP injectables) or prefer topical protection (e.g., RMs). Regardless of the mode of administration, the effectiveness of these HIV prevention strategies will require that users have access to safe and acceptable products which are easy to use consistently. Our trial addresses the limited data available regarding the safety and acceptability of and adherence to enemas, inserts, and suppositories as potential modalities through which to deliver a rectal microbicide. findings from this trial demonstrate high safety profiles, alongside high levels of acceptability and adherence, among all three modalities. future research examining the acceptability, adherence, safety, and efficacy of promising prep candidates using these three rectal microbicide modalities is encouraged.

## Supporting information

**S1 Checklist. CONSORT checklist for MTN-035 trial.**
(DOC)

**S1 File. MTN-035 protocol.**
(PDF)

## Acknowledgments

The study team gratefully acknowledges the study participants of MTN-035. We are grateful to the local research teams for their work. We also recognize the contributions of staff across the study sites. In Malawi, we recognize the work of Abigail Mnemba, Alinafe Kamanga, Annie Munthali, Daniel Gondwe, Linly Seyama, Yamikani Mbilizi, Noel Kayange, Mary Chadza, and Josiah Mayani. In South Africa, the MTN-035 team included Helen Rees, Kerushini Moodley, Krishnaveni (Krina) Reddy, Thesla Palanee-Phillips, Andile Twala, Ashleigh Jacques, Tsitsi Nyamuzihwa, Nazneen Cassim. In Peru, we recognize the work of Ana Miranda, Diana Morales, Helen Chapa, Javier Valencia, Milagros Sabaduche, Pedro Gonzales, Karina Pareja, Katherine Milagros, and Charri Macassi. We also recognize the Thailand MTN-035 team, including the work by Suwat Chariyalertsak, Pongpun Saokhieo, Veruree Manoyos, Nataporn Kosachunhanan, and Piyathida Sroysuwan. In the United States, we recognize the work of Allison

Matthews, Amy Player, Andrea Thurman, Carol Mitchell, Christine O'Neill, Christy Pappalardo, Christopher Quan, Cindy Jacobson, Clifford Yip, Craig Hendrix, Craig Hoesley, Danielle Camp, Deon Powell, Devika Singh, Diana Ng, Edward Livant, Elizabeth Brown, Emily Helms, Emily Schaeffer, Faye Heard, Gina Brown, Gustavo Doncel, Holly Gundacker, Hyman Scott, Jackie Fitzpatrick, James Gavel, Jeanna Piper, Jenna Weber, Jennifer Schille, Jessica Webster, Jessica Maitz, Jillian Zemanek, Jim Pickett, Jonathan Lucas, Julie Nowak, Kathleen Dietz, Ken Ho, Krissa Welch, Kristine Heath, Lisa Rohan, Lizardo Lacanlale, Lynn Mitterer, Lorna Richards, Marcus Bolton, Mei Song, Naana Cleland, Nicholas Ng, Nicole Macagna, Nnennaya Okey-Igwe, Onkar Singh, Patricia Peters, Rebecca Giguere, Renee Weinman, Roberta Black, Scott Fields, Sharon Riddler, Sharon Hillier, Sherri Karas, Sherri Johnson, Stacey Edick, Sufia Dadabhai, Susan Buchbinder, Taha Taha, Tarana Billups, Teri Senn, Theresa Wagner, Tim McCormick, and Yuqing Jiao. The rectal placebo inserts used in this study were provided by CONRAD as part of a project entitled Development of Novel On-Demand and Longer-Acting Microbicide Product Leads funded by a cooperative agreement between the US Agency for International Development (USAID) and Eastern Virginia Medical School (AID-OAA-A-14-00010).

## Author Contributions

**Conceptualization:** Jose A. Bauermeister, Yuqing Jiao, Elizabeth Brown, Jonathan Lucas, Cindy Jacobson, Devika Singh, Ken Ho, Albert Liu, Jeanna Piper.

**Data curation:** Clara Dominguez Islas, Yuqing Jiao, Jillian Zemanek, Devika Singh.

**Formal analysis:** Jose A. Bauermeister, Clara Dominguez Islas, Yuqing Jiao, Elizabeth Brown, Edward Livant.

**Funding acquisition:** Jeanna Piper.

**Investigation:** Jose A. Bauermeister, Clara Dominguez Islas, Ryan Tingler, Elizabeth Brown, Jillian Zemanek, Rebecca Giguere, Ivan Balan, Sherri Johnson, Nicole Macagna, Jonathan Lucas, Matthew Rose, Cindy Jacobson, Edward Livant, Devika Singh, Ken Ho, Craig Hoesley, Albert Liu, Noel Kayange, Thesla Palanee-Phillips, Suwat Chariyalertsak, Pedro Gonzales, Jeanna Piper.

**Methodology:** Jose A. Bauermeister, Clara Dominguez Islas, Yuqing Jiao, Ryan Tingler, Elizabeth Brown, Jillian Zemanek, Ivan Balan, Sherri Johnson, Nicole Macagna, Cindy Jacobson, Edward Livant, Devika Singh, Ken Ho, Craig Hoesley, Albert Liu, Thesla Palanee-Phillips, Jeanna Piper.

**Project administration:** Jose A. Bauermeister, Ryan Tingler, Elizabeth Brown, Jillian Zemanek, Rebecca Giguere, Ivan Balan, Sherri Johnson, Nicole Macagna, Jonathan Lucas, Matthew Rose, Cindy Jacobson, Clare Collins, Edward Livant, Devika Singh, Ken Ho, Craig Hoesley, Albert Liu, Noel Kayange, Thesla Palanee-Phillips, Suwat Chariyalertsak, Pedro Gonzales, Jeanna Piper.

**Resources:** Sherri Johnson, Nicole Macagna, Jonathan Lucas, Matthew Rose, Cindy Jacobson, Clare Collins, Devika Singh, Ken Ho, Thesla Palanee-Phillips, Suwat Chariyalertsak, Pedro Gonzales, Jeanna Piper.

**Supervision:** Elizabeth Brown, Jillian Zemanek, Ivan Balan, Sherri Johnson, Nicole Macagna, Devika Singh, Ken Ho, Albert Liu, Noel Kayange, Pedro Gonzales, Jeanna Piper.

**Validation:** Clara Dominguez Islas, Yuqing Jiao, Elizabeth Brown, Devika Singh, Jeanna Piper.

**Visualization:** Yuqing Jiao, Elizabeth Brown.

**Writing – original draft:** Jose A. Bauermeister, Clara Dominguez Islas, Yuqing Jiao, Elizabeth Brown, Jillian Zemanek, Sherri Johnson, Nicole Macagna.

**Writing – review & editing:** Jose A. Bauermeister, Clara Dominguez Islas, Yuqing Jiao, Ryan Tingler, Elizabeth Brown, Jillian Zemanek, Rebecca Giguere, Ivan Balan, Sherri Johnson, Nicole Macagna, Jonathan Lucas, Matthew Rose, Cindy Jacobson, Clare Collins, Edward Livant, Devika Singh, Ken Ho, Craig Hoesley, Albert Liu, Noel Kayange, Thesla Palanee-Phillips, Suwat Chariyalertsak, Pedro Gonzales, Jeanna Piper.

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
