## [Decision Letter · Decision Letter 0]

12 Jan 2023

PONE-D-22-27511A Randomized Trial of Safety, Acceptability and Adherence of Three Rectal Microbicide Placebo Formulations among Young Sexual and Gender Minorities Who Engage in Receptive Anal Intercourse (MTN-035)PLOS ONE

Dear Dr. Bauermeister,

Thank you for submitting your manuscript to PLOS ONE. After careful consideration, we feel that it has merit but does not fully meet PLOS ONE’s publication criteria as it currently stands. Therefore, we invite you to submit a revised version of the manuscript that addresses the points raised during the review process.

The manuscript has been evaluated by three reviewers, and their comments are available below.

Overall, the reviewers have expressed their congratulations on the quality of the manuscript and have offered several suggestions to further enhance the study methodology. In particular, they have noted that the study population and eligibility criteria were not described in enough detail to allow for replication of the study. As a result, the reviewers have raised concerns about the generalizability of the results. The full details of their comments may be seen below.

Could you please revise the manuscript to carefully address the concerns raised?

We look forward to receiving your revised manuscript.

Kind regards,

Lucinda Shen, MSc

Staff Editor

PLOS ONE

Journal Requirements:

“The study was designed and implemented by the Microbicide Trials Network (MTN). From 2006 until November 30, 2021, the MTN was an HIV/AIDS clinical trial network funded by the National Institute of Allergy and Infectious Diseases (UM1AI068633, UM1AI068615, UM1AI106707), with co-funding from the Eunice Kennedy Shriver National Institute of Child Health and Human Development and the National Institute of Mental Health, all components of the U.S. National Institutes of Health. The content is solely the responsibility of the authors and does not necessarily represent the official views of the National Institutes of Health.”

“I have read the journal's policy and the authors of this manuscript have the following competing interests: Dr. Liu has received funding for investigator sponsored research projects from Gilead Sciences and Viiv Healthcare.  Gilead Sciences has donated study drug to studies led by Dr. Liu.”

Reviewers' comments:

Reviewer's Responses to Questions

**Comments to the Author**

1. Is the manuscript technically sound, and do the data support the conclusions?

Reviewer #1: Yes

Reviewer #2: Yes

Reviewer #3: Yes

2. Has the statistical analysis been performed appropriately and rigorously? 

Reviewer #1: Yes

Reviewer #2: Yes

Reviewer #3: Yes

3. Have the authors made all data underlying the findings in their manuscript fully available?

Reviewer #1: Yes

Reviewer #2: No

Reviewer #3: No

4. Is the manuscript presented in an intelligible fashion and written in standard English?

Reviewer #1: Yes

Reviewer #2: Yes

Reviewer #3: Yes

5. Review Comments to the Author

Reviewer #1: This paper summarized the results of a RCT to assess acceptability and adherence to three methods for self-administering rectal microbicide PrEP for the prevention of HIV transmission via receptive anal sex. The paper is thorough in its description of methods and well written. I mostly have minor comments and points of clarification.

Methods

Eligibility criteria: What does "ability and willingness to provide adequate locator information" mean?

Eligibility criteria: How was "general good health" assessed?

This is more editorial, but there are a number of unusual acronyms (e.g., RM, PUEV) that reduce readability.

Results

Page 14: First paragraph under Study Product Use Period Completion: In one instance n = 1 is reported as 1%, in another n = 1 is reported as <1%.

Logistic mixed model results: There is a substantial difference in acceptability in the methods based on the point estimates here that is not clear in the raw estimates of acceptability. The authors address possible reasons for differences in acceptability by site in the discussion, but the models account for site. The confidence intervals are relatively wide, as well. Was the study powered for these analyses? Additional discussion about these points is warranted.

Reviewer #2: This study aimed to evaluate the safety, acceptability and adherence of three rectal microbicide placebo formulations for young sexual and gender minorities who engage in receptive anal intercourse. It is important to develop effective products such as these to complement currently available products such as oral PrEP and condoms. The paper is well written and detailed. However I do have some issues to raise.

Major comments

1. It would help to have more detailed characterisation of the study population. The cross-over procedure is explained in great detail, but the population is referred to as “sexual and gender minorities” without really explaining this, and the baseline characteristics table does not give sufficient detail either, only including gender and sex at birth, but not information such as MSM status or frequency of RAI practice.

2. The compliance is extremely high, given the number of visits required, the number of invasive procedures and that at least some of the study was conducted during the pandemic. I would like to see further detail of how this was achieved, and even more importantly, how generalisable are these results? I can’t imagine that the study participants are representative of all those who practise RAI, or even “sexual and general minorities” populations specifically. I imagine acceptability and adherence would be considerably lower in populations less able to commit to the study schedule. Related to this, I feel that the recruitment procedure is skimmed over. How were individuals currently practising RAI targeted? So few individuals were excluded because they did not fulfil the inclusion criteria. One inclusion criterion is “a reported history of consensual RAI at least three times in the past three months”? How was that ascertained without administering some kind of pre-recruitment questionnaire as part of the screening process?

3. It’s important to set out differences in results by setting more clearly. The authors provide odds ratios in Table 5 but no acceptability/adherence percentages stratified by setting. It’s important to understand how much acceptability and adherence vary by setting, so we have an idea how generalisable results may be between settings for future studies.

4. The abstract doesn’t seem to talk about any differences in outcomes between the three products, even though the results showed that the enema had a higher adherence than the other products.

5. Limitations section of Discussion: I think this should be extended. Following on from point 3 re generalisability, I think authors should comment on other populations practising RAI such as heterosexual women. The study population here was almost universally individuals born male. Participants were instructed on how to use the products and first used them in the clinic – is there any comment on potential for misuse of products if users are provided with written instructions only/guidance videos etc? Authors talk about “real-world acceptability” but the conditions in this study are very far from how such products would be used in a real-world situation. Are they really reaching the “intended end-users”? Authors mention social desirability bias but was there risk of recall bias (top page 12: “Thinking about your experience during these past four weeks, in how many of the weeks did you miss a rectal [study product] application?”?

6. Any study such as this, evaluating acceptability, cries out for a mixed methods design. This study really suffers from having no qualitative research component to start to understand and interpret the quantitative outcomes. Additionally, at the many clinic visits, there would have been opportunity to ask pertinent closed and open-ended questions. Instead, there is quite a lot of conjecture in the Discussion which could have been answered with better questionnaire design e.g., enema use and user’s practices prior to RAI. Authors could have asked participants about these practices. Authors also mention that San Francisco-based participants may already have access to other PrEP products which may be responsible for the differences in acceptability by setting. One of the study questionnaires could have asked about this. This weakness needs to be acknowledged in the Limitations section.

7. I wanted to see a lot more detail on RAI activity, including how many weeks participants reported no RAI. I’d have liked to see this in a table with frequency of RAI activity by baseline characteristics.

Minor comments

1. There are quite a few abbreviations that I wouldn’t consider necessary such as AEs, RMs, SGM and PUEVs.

2. Figure 1 refers to “douche” rather than “enema”. It also seems overly complicated.

3. Abstract: it may be clearer if the abstract was structured (IMRAD format). “204 adverse events were reported by 98 participants” – provide percentage of participants reporting at least one adverse event.

4. “The efficacy and effectiveness of emerging HIV prevention drug depends” – shouldn’t this be “HIV prevention products depend”?

5. Introduction line 4: provide references for HIV prevalence by region statistics.

6. Introduction paragraph 1 last sentence: shouldn’t it be “drugs targeting multiple STIs” rather than “multiple drugs”?

7. Page 5 last sentence: any references at all for the “few studies with SGM populations have examined the acceptability, uptake of, and adherence to inserts, suppositories, or enemas”?

8. The countries included should be stated in the abstract.

9. First line of page 7: this should really be in the Results section.

10. Figure 1 seems overly detailed, and there’s a lot of overlap with Table 1 – they can probably be combined. I think it should show the 1 week wash-out period in some way.

11. Jacobson et al are mentioned (top page 11): please describe more about the overlap between papers and what further information Jacobson et al provides that would be useful to the reader when interpreting this paper.

12. Adherence per sex act (page 16): shouldn’t this be referring to “per RAI act” rather than just “per sex act”?

13. Discussion Strengths and Limitations paragraph 2: provide percentages for adherence per protocol and per sex act, and then the percentages for hypothetical acceptability reported by all the other studies that are cited, for comparison.

14. Page 18 first sentence: “users’ behavioural practices prior to sex” – should be “prior to RAI”. Authors should be careful about these distinctions throughout the manuscript.

15. Table 2 row headers: add country for each setting.

16. Table 2 states that 100% of Blantyre participants were “Other African tribe” ethnicity. Surely, as 100% of participants from this site were from this one group, this could be more specific?

17. Table 4 label “95% Confidence Interval” doesn’t seem quite right, as there’s a central estimate then 95%CI in brackets.

Reviewer #3: This is a generally well-conducted study. There are some aspects of the reporting that could be improved. In particular, I would recommend that the authors adhere to the CONSORT guidelines. For example, the first paragraph under "methods" reports results, which should be in the results section. Instead they could have began by describing the study design, followed by eligibility, study procedures, and outcomes. One key element which is completely omitted is how the study sample size was determined - this should be reported. The statistical analysis seems mostly okay; one minor suggestion is to report standard errors instead of standard deviations in the analysis of "adherence per-sex-act" (but keep standard deviations elsewhere where they have been reported for descriptive purposes).

6. PLOS authors have the option to publish the peer review history of their article (what does this mean?). If published, this will include your full peer review and any attached files.

Reviewer #1: No

Reviewer #2: No

Reviewer #3: No

---

## [Author Response · Author response to Decision Letter 0]

15 Feb 2023

Reviewer 1

This paper summarized the results of a RCT to assess acceptability and adherence to three methods for self-administering rectal microbicide PrEP for the prevention of HIV transmission via receptive anal sex. The paper is thorough in its description of methods and well written. I mostly have minor comments and points of clarification.

• Thank you. We have clarified the points raised below.

Eligibility criteria. What does "ability and willingness to provide adequate locator information" mean? How was "general good health" assessed?

• We appreciate the Reviewer catching the typo on our inclusion criteria. We now clarify that participants had to be able and willing to provide “adequate contact and location information”. We also clarify that they had to be “deemed in general good health by a healthcare provider at Screening and Enrollment”.

Acronyms. This is more editorial, but there are a number of unusual acronyms (e.g., RM, PUEV) that reduce readability.

• Thank you. We have reduced abbreviations (i.e., RM and PUEV) to improve readability wherever possible.

Prevalence. Page 14: First paragraph under Study Product Use Period Completion: In one instance n = 1 is reported as 1%, in another n = 1 is reported as <1%.

• Thank you. We should have used <1%. This was updated in our revised manuscript. 

Logistic mixed model results: There is a substantial difference in acceptability in the methods based on the point estimates here that is not clear in the raw estimates of acceptability. The authors address possible reasons for differences in acceptability by site in the discussion, but the models account for site. The confidence intervals are relatively wide, as well. Was the study powered for these analyses? Additional discussion about these points is warranted.

• We would like to address the apparent difference in acceptability from the logistic model relative to raw estimates. Although raw estimates of the Odds Ratios (OR) are not provided in the manuscript, they are in fact not very different from those obtained from the adjust logistic model: 

OR(Insert vs Enema) = [0.72/(1-0.72)]/[0.73/(1-0.73)] = 0.95 (Adjusted OR = 0.97)

OR(Suppositories vs Enema) = [0.66/(1-0.66)]/[0.73/(1-0.73)] = 0.72 (Adjusted OR = 0.69)

We hope that the reviewer will agree that is the use of a different scale (odds ratio) what gives the impression of a more substantial difference between some of the modalities, rather than it being a consequence of including adjustment of other factors in the logistic model. That said, we would like to clarify that the principal objective of the study was not to compare the different modalities, but to evaluate the acceptability and adherence of each one and to provide estimates. While the sample size of the study allowed for high power for ruling out acceptability/adherence rates below 70%, the study was not powered for formal comparisons between modalities. The logistic regression model analysis was performed to test for any potential differences by randomize sequence (to check for possible effect of the order in which participants used the products, or possible carry-over effect), and (ii) to test for the differences observed between sites. While we think it reasonable to include the study product in these models (which in turns allowed us to make some statements regarding some of the differences being statistically significant), we acknowledge that failing to declare some of the differences as significant does not imply the absence of such differences. To avoid confusion, the Statistical Analysis section of the Methods has been updated to clarify the main objective of the study. We have also removed some statements previously included in the Results section, which probably unduly emphasized these statistical tests.

Reviewer #2

This study aimed to evaluate the safety, acceptability and adherence of three rectal microbicide placebo formulations for young sexual and gender minorities who engage in receptive anal intercourse. It is important to develop effective products such as these to complement currently available products such as oral PrEP and condoms. The paper is well written and detailed. However I do have some issues to raise.

• Thank you for your review. We respond to the points raised below, as well as in the manuscript.

Description of the study population. It would help to have more detailed characterisation of the study population. The cross-over procedure is explained in great detail, but the population is referred to as “sexual and gender minorities” without really explaining this, and the baseline characteristics table does not give sufficient detail either, only including gender and sex at birth, but not information such as MSM status or frequency of RAI practice.

• We have now provided a definition of sexual gender minorities (i.e., populations with same-sex or same-gender attractions or behaviors and who may identify with a non-heterosexual identity such as gay, bisexual, queer, etc.) in the Introduction.

• Thank you. We note that the sample’s baseline sexual behaviors as part of their Inclusion criteria (i.e., individuals self-report engaging in consensual receptive anal sex at least three times in the past three months and have the expectation to maintain at least that frequency of RAI during study participation). As requested, we have also included additional baseline information regarding participants’ sexual behaviors in the prior 30 days and prior history with the three study products in the Results:

“Participants reported having had an average of 3 male partners (SD=6.35; range: 0-70) in the prior 30 days. Participants’ average total number of RAI occasions during that 30-day period was 4.76 (SD=8.50; range: 0-100), with an average of 2.57 (SD=7.74; range: 0-100) condomless RAI occasions self-reported during the same period. Two thirds of participants (n=142; 65.4%) reported prior use of an enema, with fewer participants self-reporting that they had used a suppository (n=8; 3.7%) or insert (n=10; 4.6%) prior to RAI.”

Generalizability of results. The compliance is extremely high, given the number of visits required, the number of invasive procedures and that at least some of the study was conducted during the pandemic. I would like to see further detail of how this was achieved, and even more importantly, how generalisable are these results? I can’t imagine that the study participants are representative of all those who practise RAI, or even “sexual and general minorities” populations specifically. I imagine acceptability and adherence would be considerably lower in populations less able to commit to the study schedule. 

• We agree that participants in a clinical trial may not be representative of the general population. We have expanded the limitations section to highlight this point. Specifically, we now state: “Second, we recruited a convenience sample of participants willing to use each of the study products at least once per week, as required by the protocol. We acknowledge that the generalizability of our clinical trial findings may not be representative of all individuals practicing RAI.”

Recruitment and screening. Related to this, I feel that the recruitment procedure is skimmed over. How were individuals currently practising RAI targeted? So few individuals were excluded because they did not fulfil the inclusion criteria. One inclusion criterion is “a reported history of consensual RAI at least three times in the past three months”? How was that ascertained without administering some kind of pre-recruitment questionnaire as part of the screening process?

• Thank you. We have expanded the recruitment paragraph to highlight the procedures that different sites undertook. We now state: Participants were recruited from a variety of sources, including outpatient clinics, universities, community-based locations, online websites, and social networking applications. In addition, participants were also referred to the study from other local research projects, research registries and other health and social service providers. At some sites, prospective participants were pre-screened by phone, using an IRB-approved phone script, to assess presumptive eligibility based on select behavioral and medical eligibility requirements. This includes a review of the prospective participants’ sexual history and engagement in receptive anal sex in their lifetime and within the previous three months. For those deemed presumably eligible when a phone screen was conducted, a screening visit was scheduled. A re- affirmation of all eligibility criteria was obtained and confirmed during a formal screening and enrollment visit, described below.

Differences in results. It’s important to set out differences in results by setting more clearly. The authors provide odds ratios in Table 5 but no acceptability/adherence percentages stratified by setting. It’s important to understand how much acceptability and adherence vary by setting, so we have an idea how generalisable results may be between settings for future studies.

• Thank you. We have added results by-site in Table 4.

Abstract. The abstract doesn’t seem to talk about any differences in outcomes between the three products, even though the results showed that the enema had a higher adherence than the other products.

• Thank you. We would like to clarify that the principal objective of the study was not to compare the different modalities, but to evaluate the acceptability and adherence of each one and to provide estimates. While the sample size of the study allowed for high power for ruling out acceptability/adherence rates below 70%, the study was not powered for formal comparisons between modalities. Therefore, we have not included this detail in the abstracts to avoid misdirecting the reader.

Limitations section of Discussion: I think this should be extended. Following on from point 3 re generalisability, I think authors should comment on other populations practising RAI such as heterosexual women. The study population here was almost universally individuals born male. Participants were instructed on how to use the products and first used them in the clinic – is there any comment on potential for misuse of products if users are provided with written instructions only/guidance videos etc? Authors talk about “real-world acceptability” but the conditions in this study are very far from how such products would be used in a real-world situation. Are they really reaching the “intended end-users”? Authors mention social desirability bias but was there risk of recall bias (top page 12: “Thinking about your experience during these past four weeks, in how many of the weeks did you miss a rectal [study product] application?”?

• Thank you for these comments. We have amended the Limitations section to note the potential for recall bias: “Second, there is a possibility of recall bias when participants completed their surveys.”. We have also made a note regarding the need for future research outside of a clinical trial design: “Finally, while we designed our clinical trial to resemble participants’ product use to as close as ‘real-world’ settings possible while maintaining rigor, we acknowledge that the trial protocols may hinder the social validity of the findings. Future research examining these products in real-world situations may further clarify their potential for use as rectal microbicide modalities.”

• While we agree that it is important to scientifically study the sexual practices of other populations (e.g., heterosexual women who engage in RAI) and their potential for rectal microbicides, we have chosen not to make reference to this group for two reasons: (1) we want to centralize the experience of sexual and gender minorities, and (2) heterosexual women’s sociocultural and behavioral correlates to RAI are unique and may not be parallel to the experience of sexual and gender minorities.

Mixed-methods design. Any study such as this, evaluating acceptability, cries out for a mixed methods design. This study really suffers from having no qualitative research component to start to understand and interpret the quantitative outcomes. Additionally, at the many clinic visits, there would have been opportunity to ask pertinent closed and open-ended questions. Instead, there is quite a lot of conjecture in the Discussion which could have been answered with better questionnaire design e.g., enema use and user’s practices prior to RAI. Authors could have asked participants about these practices. Authors also mention that San Francisco-based participants may already have access to other PrEP products which may be responsible for the differences in acceptability by setting. One of the study questionnaires could have asked about this. This weakness needs to be acknowledged in the Limitations section.

• Thank you for your suggestions. The objective of this paper is to present the primary trial outcomes data for MTN 035. We collected qualitative data as part of MTN 035, yet these data will be presented elsewhere as they inform secondary and exploratory outcomes. It is beyond the scope of this manuscript to include all these data.

RAI. I wanted to see a lot more detail on RAI activity, including how many weeks participants reported no RAI. I’d have liked to see this in a table with frequency of RAI activity by baseline characteristics.

• Although we do not explicitly provide the number of participants that reported no RAI during the study product use periods, the summaries of adherence per-sex-act are calculated for participants who reported at least one sex act, with those numbers provided along with the summaries. Among participants who reported at least one sex act during the product use period, the percentages of participants fully adherent per RAI-act were similar among the three study products: insert (n=99/168; 58.9%), suppository (n=101/174; 58.0%) and enema (n=107/182; 58.8%). 

Abbreviations. There are quite a few abbreviations that I wouldn’t consider necessary such as AEs, RMs, SGM and PUEVs.

• Thank you. We have reduced abbreviations to improve readability wherever possible.

Abstract. It may be clearer if the abstract was structured (IMRAD format). “204 adverse events were reported by 98 participants” – provide percentage of participants reporting at least one adverse event. 8. The countries included should be stated in the abstract.

• Thank you. These edits are now included.

Grammatical edits and recommendations. “The efficacy and effectiveness of emerging HIV prevention drug depends” – shouldn’t this be “HIV prevention products depend”? Introduction line 4: provide references for HIV prevalence by region statistics. Introduction paragraph 1 last sentence: shouldn’t it be “drugs targeting multiple STIs” rather than “multiple drugs”? 

• Thank you. The requested edits have been made 

References. Page 5 last sentence: any references at all for the “few studies with SGM populations have examined the acceptability, uptake of, and adherence to inserts, suppositories, or enemas”?

• The sentence focuses on acceptability and use prior to sex. We now of one study by Carballo-Dieguez et al. (citation 27) which did so and is already included in the text.

Figure 1 and Table 1. Figure 1 seems overly detailed, and there’s a lot of overlap with Table 1 – they can probably be combined. I think it should show the 1 week wash-out period in some way.

• We respectfully disagree. To ensure clarity, we have left Table 1 as the descriptor of the randomized cross-over design as an illustration. Figure 1 denotes the in-depth detail required for the CONSORT diagram. 

Jacobson paper. Jacobson et al are mentioned (top page 11): please describe more about the overlap between papers and what further information Jacobson et al provides that would be useful to the reader when interpreting this paper.

• There is no significant overlap between the Jacobson paper and the current trial outcomes paper. The Jacobson et al. paper describes how pharmacists responded to site-specific COVID-19 protocols to minimize disruptions to the trial. 

Clarity regarding “per RAI act vs per sex act”. Adherence per sex act (page 16): shouldn’t this be referring to “per RAI act” rather than just “per sex act”? Page 18 first sentence: “users’ behavioural practices prior to sex” – should be “prior to RAI”. Authors should be careful about these distinctions throughout the manuscript.

• Clarification has been made throughout the manuscript.

Headers in Table 2. Table 2 row headers: add country for each setting.

• Added as requested.

Ethnic distribution of Blantyre participants. Table 2 states that 100% of Blantyre participants were “Other African tribe” ethnicity. Surely, as 100% of participants from this site were from this one group, this could be more specific?

• Unfortunately, no further information about ethnicity of this participants was collected on case report forms.

OR and CI labels. “95% Confidence Interval” doesn’t seem quite right, as there’s a central estimate then 95%CI in brackets.

• We have clarified the labels accordingly to note Odds Ratio estimates, followed by lower and upper confidence intervals in our tables.

Reviewer #3

This is a generally well-conducted study. There are some aspects of the reporting that could be improved. In particular, I would recommend that the authors adhere to the CONSORT guidelines. For example, the first paragraph under "methods" reports results, which should be in the results section. Instead they could have began by describing the study design, followed by eligibility, study procedures, and outcomes. 

• Thank you; we have made the appropriate changes to align with the CONSORT guideline.

Sample size. One key element which is completely omitted is how the study sample size was determined - this should be reported. The statistical analysis seems mostly okay; one minor suggestion is to report standard errors instead of standard deviations in the analysis of "adherence per-sex-act" (but keep standard deviations elsewhere where they have been reported for descriptive purposes).

• We thank the reviewer for his comment. Additional information about the sample size has been provided in the corresponding section. Also, standard errors (SEs) have been provided instead of standard deviations (SDs) when describing the reported average of sex acts.

---

## [Decision Letter · Decision Letter 1]

14 Mar 2023

PONE-D-22-27511R1A Randomized Trial of Safety, Acceptability and Adherence of Three Rectal Microbicide Placebo Formulations among Young Sexual and Gender Minorities Who Engage in Receptive Anal Intercourse (MTN-035)PLOS ONE

Dear Dr. Bauermeister,

Thank you for submitting your manuscript to PLOS ONE. After careful consideration, we feel that it has merit but does not fully meet PLOS ONE’s publication criteria as it currently stands. Therefore, we invite you to submit a revised version of the manuscript that addresses the points raised during the review process. Please see 2 remaining comments from reviewers that need to be addressed.

We look forward to receiving your revised manuscript.

Kind regards,

Renee Ridzon

Academic Editor

PLOS ONE

Journal Requirements:

Reviewers' comments:

Reviewer's Responses to Questions

**Comments to the Author**

1. If the authors have adequately addressed your comments raised in a previous round of review and you feel that this manuscript is now acceptable for publication, you may indicate that here to bypass the “Comments to the Author” section, enter your conflict of interest statement in the “Confidential to Editor” section, and submit your "Accept" recommendation.

Reviewer #1: All comments have been addressed

Reviewer #2: (No Response)

Reviewer #3: All comments have been addressed

2. Is the manuscript technically sound, and do the data support the conclusions?

Reviewer #1: Yes

Reviewer #2: Yes

Reviewer #3: (No Response)

3. Has the statistical analysis been performed appropriately and rigorously? 

Reviewer #1: Yes

Reviewer #2: Yes

Reviewer #3: (No Response)

4. Have the authors made all data underlying the findings in their manuscript fully available?

Reviewer #1: Yes

Reviewer #2: No

Reviewer #3: (No Response)

5. Is the manuscript presented in an intelligible fashion and written in standard English?

Reviewer #1: Yes

Reviewer #2: Yes

Reviewer #3: (No Response)

6. Review Comments to the Author

Reviewer #1: (No Response)

Reviewer #2: The authors appear to have responded to reviewers’ comments satisfactorily, although my comments have been rearranged and paraphrased so it was difficult to keep track of authors’ responses.

Table 4 “Odds Ratio and 95% Confidence Interval” row looks like it's the percentage with high acceptability with missing data removed, with 95%CI i.e. not an odds ratio. I strongly agree that acceptability 95%CI should be quoted, especially now the authors have made it clear that 70% acceptability is an important benchmark. However, it seems odd to give the % Yes at the top of that table, which includes those with missing data in the denominator, then a few rows down present a very slightly different estimate (because missing data are removed). So the table needs correct labelling and clearer signposting as to which statistic (with or without those with missing data) is the most important outcome. Maybe just state number with missing data as a footnote and present the percentages excluding missing data only.

Reviewer #3: All through the article, beginning with the abstract and all over the results, please indicate the confidence intervals reported. This can be as simple as changing 'CI' to '95%CI' if indeed these are 95% confidence intervals.

7. PLOS authors have the option to publish the peer review history of their article (what does this mean?). If published, this will include your full peer review and any attached files.

Reviewer #1: No

Reviewer #2: No

Reviewer #3: No

---

## [Author Response · Author response to Decision Letter 1]

17 Mar 2023

PONE-D-22-27511R1

A Randomized Trial of Safety, Acceptability and Adherence of Three Rectal Microbicide Placebo Formulations among Young Sexual and Gender Minorities Who Engage in Receptive Anal Intercourse (MTN-035)

PLOS ONE

Thank you for the opportunity to revise our manuscript under consideration in PLOS ONE. We have addressed the remaining questions/comments from the Reviewers, and uploaded both a version with track changes and an unmarked version of the manuscript with the requested revisions.

Reviewer #2: 

The authors appear to have responded to reviewers’ comments satisfactorily, although my comments have been rearranged and paraphrased so it was difficult to keep track of authors’ responses.

Table 4 “Odds Ratio and 95% Confidence Interval” row looks like it's the percentage with high acceptability with missing data removed, with 95%CI i.e. not an odds ratio. I strongly agree that acceptability 95%CI should be quoted, especially now the authors have made it clear that 70% acceptability is an important benchmark. However, it seems odd to give the % Yes at the top of that table, which includes those with missing data in the denominator, then a few rows down present a very slightly different estimate (because missing data are removed). So the table needs correct labelling and clearer signposting as to which statistic (with or without those with missing data) is the most important outcome. Maybe just state number with missing data as a footnote and present the percentages excluding missing data only.

• Thank you. We have removed the mislabeling in our row stating Odds Ratio and now clarify that it’s the percent of participants with high acceptability/adherence. 

• We have deleted the recommended rows noting missingness and added a Note instead that states:

o Notes. Estimates exclude missing data for acceptability (Rectal insert (n=3); Suppository (n=2); Enema (n=5)) and adherence (Rectal insert (n=3); Suppository (n=3); Enema (n=2)).

Reviewer #3: 

All through the article, beginning with the abstract and all over the results, please indicate the confidence intervals reported. This can be as simple as changing 'CI' to '95%CI' if indeed these are 95% confidence intervals.

• Thank you. We have now edited the text to document that we are referring to 95% Confidence Intervals.

---

## [Editor Report · Decision Letter 2]

29 Mar 2023

A Randomized Trial of Safety, Acceptability and Adherence of Three Rectal Microbicide Placebo Formulations among Young Sexual and Gender Minorities Who Engage in Receptive Anal Intercourse (MTN-035)

PONE-D-22-27511R2

Dear Dr. Bauermeister,

We’re pleased to inform you that your manuscript has been judged scientifically suitable for publication and will be formally accepted for publication once it meets all outstanding technical requirements.

Kind regards,

Renee Ridzon

Academic Editor

PLOS ONE
---

## [Editor Report · Acceptance letter]

3 Apr 2023

PONE-D-22-27511R2 

A Randomized Trial of Safety, Acceptability and Adherence of Three Rectal Microbicide Placebo Formulations among Young Sexual and Gender Minorities Who Engage in Receptive Anal Intercourse (MTN-035) 

Dear Dr. Bauermeister:

I'm pleased to inform you that your manuscript has been deemed suitable for publication in PLOS ONE. Congratulations! Your manuscript is now with our production department. 

Kind regards, 

on behalf of

Dr. Renee Ridzon 

Academic Editor

PLOS ONE